# Object Detection of Small Insects in Time-Lapse Camera Recordings

**DOI:** 10.3390/s23167242

**Published:** 2023-08-18

**Authors:** Kim Bjerge, Carsten Eie Frigaard, Henrik Karstoft

**Affiliations:** Department of Electrical and Computer Engineering, Aarhus University, 8200 Aarhus N, Denmarkhka@ece.au.dk (H.K.)

**Keywords:** camera recording, deep learning, insect dataset, motion enhancement, object detection

## Abstract

As pollinators, insects play a crucial role in ecosystem management and world food production. However, insect populations are declining, necessitating efficient insect monitoring methods. Existing methods analyze video or time-lapse images of insects in nature, but analysis is challenging as insects are small objects in complex and dynamic natural vegetation scenes. In this work, we provide a dataset of primarily honeybees visiting three different plant species during two months of the summer. The dataset consists of 107,387 annotated time-lapse images from multiple cameras, including 9423 annotated insects. We present a method for detecting insects in time-lapse RGB images, which consists of a two-step process. Firstly, the time-lapse RGB images are preprocessed to enhance insects in the images. This motion-informed enhancement technique uses motion and colors to enhance insects in images. Secondly, the enhanced images are subsequently fed into a convolutional neural network (CNN) object detector. The method improves on the deep learning object detectors You Only Look Once (YOLO) and faster region-based CNN (Faster R-CNN). Using motion-informed enhancement, the YOLO detector improves the average micro *F*1-score from 0.49 to 0.71, and the Faster R-CNN detector improves the average micro *F*1-score from 0.32 to 0.56. Our dataset and proposed method provide a step forward for automating the time-lapse camera monitoring of flying insects.

## 1. Introduction

More than half of all the described species on Earth are insects; they are the most abundant group of animals and live in almost every habitat. There are multiple reports of declines in abundance, diversity, and biomass of insects all over the world [1,2,3,4]. Changes in the abundance of insects could have cascading effects on the food web. Bees, hoverflies, wasps, beetles, butterflies, and moths are important pollinators and prey for birds, frogs, and bats. Some of the most damaging pest species in agriculture and forestry are moths [5,6], and insects are known to be major factors in the world’s agricultural economy. Therefore, it is crucial to monitor insects in the context of global changes in climate and habitats.

Traditional insect-trapping methods [7] are based on the manual capturing and sampling of insects. Manual trapping is very tedious and labor intensive. Typically, insects are killed, manually counted, and classified by humans—which requires expert knowledge—and rare insect species are killed. Samples from the traps are used for penology studies and abundance and growth analyses across taxa, regions, and periods. Malaise traps [8] are large tent-like structures made of netting, meant to funnel insects into a common area. The net is then checked and emptied periodically over days or weeks. Light traps [9] are used to catch insects at night using different spectral lights to attract insects like moths. Light-attracted insects fly toward the light source, hit a surface surrounding the light, and can then be observed, recorded, sampled, or collected. All the above-mentioned trapping methods are labor intensive, since insects need to be collected and inspected manually for species categorization and counting. Many trapping methods are destructive, and rare species of insects can be killed. The largest bottleneck for traditional insect monitoring is in accessing data after trapping. Tools are needed to accelerate the time taken to collect, process, and identify specimens, and for the efficient capture of data and meta-data associated with an observation.

Automated insect camera traps and data-analyzing algorithms based on computer vision and deep learning are valuable tools for monitoring and understanding insect trends and their underlying drivers [10,11]. It is challenging to automate insect detection since insects move fast, and their environmental interactions, such as pollination events, are ephemeral. Insects also have small sizes [11,12] and may be occluded by flowers or leaves, making it hard to separate the objects of interest from natural vegetation. A particularly exciting prospect enabled by computer vision is the automated, non-invasive monitoring of insects and other small organisms in their natural environment. Here, image processing with deep learning models of insects can be applied either in real time [13] or batched, since time-lapse images can be stored and processed after collection [14,15,16,17,18].

Remote optical sensing driven by deep learning has drawn remarkable attention and achieved significant breakthroughs [19]. Remote sensing and image scene classification aim at labeling remote sensing images with a set of semantic classes based on their contents. In most cases, the datasets for small object detection in remote sensing images are inadequate [20]. Many researchers have used scene classification datasets for object detection, which have their limitations, such as large-sized objects outnumbering small objects in object categories. Accurate object detection is important in remote sensing. However, detecting small objects in low-resolution images remains a challenge, primarily because these objects have less visual information and cannot be easily distinguished from similar background regions. To resolve this problem, Wu et al. [21] proposed a small object detection network for low-resolution remote sensing images and addressed challenges similar to those of insect monitoring studied in our paper.

Convolutional neural networks (CNNs) are extensively used for object detection [22,23,24,25] in many contexts, including insect detection and species identification. CNNs for object detection predict bounding boxes around objects within the image, their class labels, and confidence scores. You Only Look Once (YOLO) [26,27] is a one-stage object detector that is popular in many applications and has been applied for insect detection [28]. Two-stage detectors, such as the faster region-based convolutional neural network (Faster R-CNN) [29], are also very common and have been adapted for small object detection [30].

Annotated datasets are essential for data-driven insect detectors. Data should include images of the insects for detection and images of the typical backgrounds where such insects may be found. Suppose an object detector is trained on one dataset. In this case, it will not necessarily have the same performance on time-lapse recordings from a new monitoring site. One false detection in a time-lapse image sequence of natural vegetation will cause multiple false detections in the subsequent stationary images [17].

### 1.1. Related Work

#### 1.1.1. Detection of Small Objects

Small object detection in low-resolution remote sensing images presents numerous challenges [31]. Targets are relatively small compared with the field of view, do not present distinct features, and are often grouped and lost in cluttered backgrounds.

Liu et al. [32] compared the performances of several leading deep learning methods for small object detection. They discussed the challenges and techniques for improving the detection of small objects. These techniques include fusing feature maps from shallow layers and deep layers to obtain essential spatial and semantic information. Another approach is a multi-scale architecture consisting of separate branches for small-, medium-, and large-scale objects, such as Darknet53 [27], which generates anchors of different scales. Usually, small objects require a high resolution and are difficult to recognize; here, spatial and temporal contextual information plays a critical role in small object detection [32,33].

A comprehensive review of recent advances in small object detection based on deep learning is provided by Tong et al. [34]. The review covers topics such as multi-scale feature learning [35], pyramid networks [36], data augmentation, training strategies, and context-based detection. Important requirements for the future are proposed: emerging small object detection datasets and benchmarks, small object detection methods, and frameworks for small object detection tasks.

#### 1.1.2. Detection in a Single Image

Detection of small objects in the spatial dimension of images has been investigated in several domains such as remote sensing [37] with single-shot or time-lapse images. For small object detection tasks, detection is very difficult since these small objects can be tightly grouped and interfere with background information.

Du et al. [38] proposed an extended network architecture based on YOLOv3 [39] for small-sized object detection with complex backgrounds. They added multi-scale convolution kernels with different receptive fields into YOLOv3 to improve the extraction of objects’ semantic features using an Inception-like architecture inspired by GoogleNet [40].

Huang et al. [41] proposed a small object detection method based on YOLOv4 [27] for chip surface defect inspection. They extended the backbone of the YOLOv4 architecture with an enhanced receptive field by adding an additional fusion output (104×104) from the cross-stage partial layer (CSP2) with a similarly extended neck.

These works focus on improving the architecture for detecting small objects but are only demonstrated on general datasets that don’t include insects and achieved only minor improvements.

#### 1.1.3. Detection in a Sequence of Images

With higher frame rates, such as those in video recordings [42], information in the temporal dimension can be used to improve the detection and tracking of moving objects [43,44]. The detection of small moving objects is an important research area with respect to flying insects, surveillance of honeybee colonies, and tracking the movement of insects. Motion-based detections consist principally of background subtraction and frame differencing. State-of-the-art methods aim to combine the approaches of both spatial appearance and motion to improve object detection. Here, CNNs consider both motion and appearance information to extract object locations [45,46].

LaLonde et al. [47] proposed ClusterNet for the detection of small cars in wide-area motion imagery. They achieved a state-of-the-art accuracy using a two-stage deep network where the second stage detects small objects using a large receptive field. However, the inputs are consecutive adjoining frames with frame rates of 0.8 fps.

Stojnić et al. [48] proposed how to track small moving honeybees recorded by unmanned aerial vehicle (UAV) videos. First, they performed background estimation and subtraction, followed by semantic segmentation using U-net [49] and thresholding of the segmented frame. Since a labeled dataset of small moving objects did not exist, they generated synthetic videos for training by adding small blob-like objects on real-world backgrounds. In a final test on real-world videos with manually annotated flying honeybees, they achieved a best average F1-score of 0.71 on three small video test sequences.

Aguilar et al. [50] studied small object detection and tracking in satellite videos of motorbikes. They used a track-by-detection approach to detect and track small moving targets by using CNN object detection and a Bayesian tracker. The first stage uses a lightweight motion-informed detection operator to obtain rough target locations. The second stage combines this information with a Faster R-CNN to refine the detection results. In addition, they adopted an online track-by-detection approach by using the probability hypothesis density (PHD) filter to convert detections into tracks.

Insect detection and tracking were proposed in [28], where images were recorded in real time with a frame rate of only 0.33 fps, and insect detection and species classification were achieved using YOLOv3, followed by a multiple object tracker using detected center points and the size of the object-bounding box. The camera system used edge computing in real time to detect and classify insects; however, this approach requires more computational power than simple time-lapse cameras.

Ratnayake et al. [51] presented an offline hybrid detection and tracking algorithm to monitor unmarked insects in a natural environment based on video recordings. The method was applied to track honeybees foraging outdoors using a dataset that included complex detailed backgrounds. However, this work was based on continuous video recording, which is a challenge for long-term in-field experiments, requiring sufficient hardware to process and store data.

### 1.2. Contribution

In-field camera recording naturally requires sufficient hardware to process and store videos with a real-time sampling frequency. This poses technical difficulties when the recording period is long and the hardware must operate without external power and network connections. In this paper, we focus on small object detection for time-lapse recordings, which requires less storage space.

There are two types of remote sensing: either where the observer is moving, such as video recordings via UAVs [48] or satellites [47,50], or where the observer is in a stationary position, such as monitoring of insects [28,51]. The background is the same when the observer is in a stationary position and moving objects differ from the background. Here, we will improve insect object detection using temporal images where moving objects differ from the background.

Annotated datasets are essential for data-driven insect detectors, and our work contributes with a new comprehensive dataset (for both training and especially testing) that is significant for small object detection of insects in time-lapse camera recordings. To our knowledge, only datasets [28] with insects for model training have been published. This dataset does not address the challenge of testing the model performance on new camera sites. Here, we provide a test dataset from seven different sites with more than 100,000 annotated time-lapse images.

We hypothesize that motion-informed enhancement in insect detection from time-lapse recordings will improve detection in the environment. In short, we summarize our contributions as follows:Provide a dataset with annotated small insects (primarily honeybees) and a comprehensive test dataset with time-lapse annotated recordings from different monitoring sites.Propose a new pipeline method to improve insect detection in the wild, built on motion-informed enhancement, YOLOv5, and Faster R-CNN with ResNet50 as the backbone.

## 2. Dataset

We provide a new, comprehensive benchmark dataset to evaluate data-driven methods for detecting small insects in a real natural environment.

Dataset images were collected using four recording units, each consisting of a Raspberry Pi 3B computer connected to two Logitech C922 HD Pro USB web cameras [52] with a resolution of 1920 × 1080 pixels. Images from the two cameras were stored in JPG format on an external 2TB USB hard disk.

A time-lapse program [53] installed on the Raspberry Pi was used to capture time-lapse images continuously with a frame rate of 30 s between images. The camera used automatic exposure to handle light variations in the wild related to clouds, shadows, and direct sun. Auto-focus was enabled to handle variations in the camera distance and orientation in relation to scenes with plants and insects. The system recorded images every day from 4:30 a.m. to 22:30 p.m., resulting in a maximum of 2160 images per camera per day.

During the period from 31 May to 5 August 2022, the camera systems were in operation in four greenhouses in Flakkebjerg, Denmark. The camera systems monitored insects visiting three different species of plants: *Trifolium prantese* (red clover), *Cakile maritima* (sea rocket), and *Malva sylvestris* (common mallow). The camera systems were moved during the recording period to ensure different flowering plants were recorded from a side or top camera view during the whole period of observation. A small beehive was placed in each greenhouse with western honeybees (*Apis mellifera*), meaning we expected primarily to monitor honeybees visiting plants.

A dataset for training and validation was created, based on recordings from six different cameras with side and top views of red clovers and sea rockets, as shown in Figure 1.

**Figure 1 sensors-23-07242-f001:**
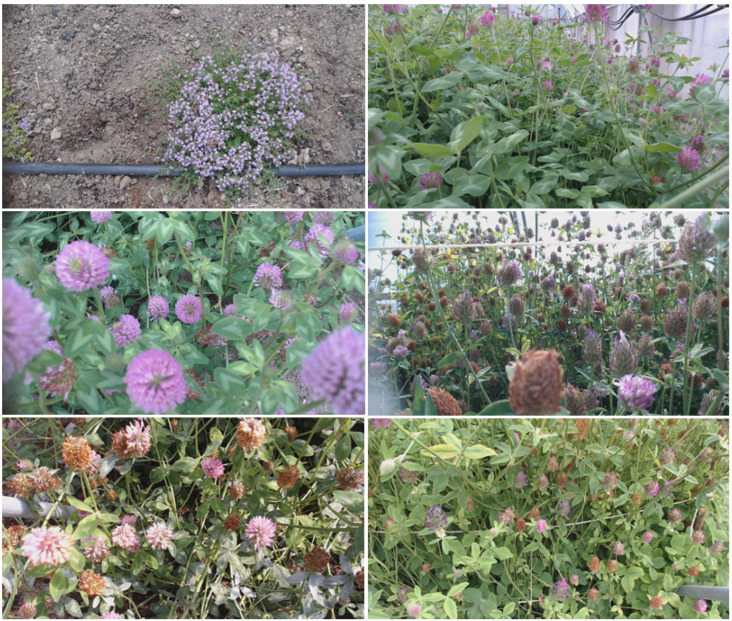
Example of six background images from camera systems monitoring flowering sea rocket and red clover plants observed from top and side views. Images were taken by camera systems at sites shown in Table 1. (S1-1 w26, S2-1 w27, S1-1 w27, S3-0 w29, S1-0 w30, and S4-1 w29).

**Table 1 sensors-23-07242-t001:** Camera sites and weeks from where data were selected to create a training and validation dataset. System number and camera Id (Sx-0/1) identify each camera. Insects are the number of annotated insects found in the selected images. Background (Back.) is the number of images without any insects where FP detections were removed. The flowering plants were observed with a top or side camera view. The plant species are sea rocket (*Cakile maritima*) and red clover (*Trifolium prantese*). Examples of background images are shown in Figure 1.

Cam.	Week	Days	Insects	Back.	View	Plant
S1-1	26	2	1079	340	Top	Rocket
S1-1	27	2	21	312	Top	Clover
S1-0	29	7	395	143	Top	Rocket
S1-0	30	7	648	115	Side	Clover
S2-1	27	7	186	136	Side	Clover
S3-0	29	7	120	308	Side	Clover
S4-0	28	7	154	533	Side	Clover
S4-0	30	7	20	468	Top	Clover
S4-1	28	7	108	77	Top	Clover
S4-1	29	7	83	93	Top	Clover
Total	10	60	2814	2525		

Finally, a comprehensive test dataset was created by selecting seven camera sites, as listed in Table 2. The test dataset was selected to have backgrounds and camera views other than those included during model training. At the seven sites, common mallow was monitored for two weeks, sea rocket was monitored for one week, and red clover was monitored for four weeks in top and side views. All images were annotated using an iterative semi-automated process using human labeling and verification of model detections to find and annotate insects in more than 100,000 images. The goal was to evaluate the object detection models on a real dataset with a distribution other than images used for training and validation.

## 3. Method

Our proposed pipeline for detecting insects in time-lapse RGB images consisted of a two-step process. In the first step, images with motion-informed enhancement (MIE) were created. In the second step, existing object detectors based on deep learning detectors used these enhanced images to improve the detection of small objects.

### 3.1. Motion-Informed Enhancement

Insects can be monitored in their natural environment with time-lapse cameras, where a time-lapse image is recorded at fixed time intervals of typically 30 or 60 s. We hypothesized that small objects in motion would be easier to detect with deep learning detectors if images also include information from the temporal dimension when training the model.

The motion-informed detection operator proposed by Aguilar et al. [50] was modified for this paper to improve insect object detection using temporal images without tracking. The detection operator estimates motion by finding the difference between consecutive frames in a time-lapse sequence. Our proposal modified this method to create an enhanced image with motion information that is used for inference and training the deep learning object detector. By using the standard RGB image format and only modifying the color content, existing object detectors can be used without modification. This approach can use popular image object detectors with CNNs, such as YOLO [26] and Faster R-CNN [29].

Three consecutive images from the time-lapse recording were used to create the enhanced image. The colored images were first converted to grayscale and blurred ($B_{k}$) with a Gaussian kernel of 5 × 5 pixels (image size: 1920 × 1080 pixels). The gray scales and blurred images were then used to create the motion likelihood, $L_{k}^{3}\left[i,j\right]$, where [i,j]∈[1..N]×[1..M] are the pixel coordinates and k∈N is the time index. This process is summarized in Equations (Equation 1) and (Equation 2).
(1)$$\Delta B_{k}\left[i,j\right]=B_{k}\left[i,j\right]-B_{k-1}\left[i,j\right]$$
(2)$$L_{k}^{3}\left[i,j\right]=|\Delta B_{k}\left[i,j\right]|+|\Delta B_{k+1}\left[i,j\right]|$$

The original colored image at time *k* was then modified to create a motion-enhanced image (*M*). Here, the enhanced blue color channel ($M_{b}$) was replaced by a combination of the original red ($I_{r}$) and blue color channels ($I_{b}$), shown in Equation (3). The motion likelihood ($L^{3}$) was inserted in the enhanced red channel, see Equation (4), and the original green channel was unchanged, and copied to the enhanced green channel (Equation (5)).
(3)$$M_{b}\left[i,j\right]=0.5I_{b}\left[i,j\right]+0.5I_{r}\left[i,j\right]$$
(4)$$M_{r}\left[i,j\right]=L_{k}^{3}\left[i,j\right]$$
(5)$$M_{g}\left[i,j\right]=I_{g}\left[i,j\right]$$

The results of the proposed method are illustrated in Figure 2 and Figure 3. The figures show how the motion information is created and is finally observed as a red color on the moving insect in the enhanced image. Most of the background colors of the leaves are green and unchanged in the enhanced image. Flower colors such as pink, red, and orange are mixed in the blue channel.

### 3.2. Object Detection with Deep Learning

Image object detection methods based on deep learning rely solely on spatial image information to extract features and detect regions of objects in the image. Object detection combines the tasks of localization by drawing a bounding box around each object of interest in the image and assigning them a class label.

One-stage object detectors predict the object with a bounding box boundary, detect whether an object is present, and classify the object in the same process stage. One-state detectors are typically faster than two-stage detectors at the cost of a lower accuracy. However, fast execution is important when millions of images need to be processed on a large scale, for example, in remote sensing. Although remarkable results have been achieved across several benchmarks, the detectors’ performances decrease with small objects in complex environments, such as those in insect monitoring. Two-stage detectors perform region proposals before inference and classification.

In our work, Faster R-CNN with a backbone of ResNet50 [54] and YOLOv5 [55] with a backbone of CSPDarknet53 were used to detect small insects in wildlife images. YOLO is a one-stage object detector and Faster R-CNN is a two-stage detector.

YOLO is a state-of-the-art, real-time object detection algorithm that uses a single neural network to process an entire image. The image is divided into regions and the algorithm predicts probabilities and bounding boxes for each region. YOLOv5 uses the modified backbone CSPDarknet53 and introduces new features such as “bag of specials” and “bag of freebies”, where advanced data augmentation for training is improved without affecting the inference cost. The CSPDarknet53 architecture uses three residual skip connections that perform detections at three different scales, including small objects.

Faster R-CNN uses a region proposal network (RPN), which is a fully convolutional network (FCN) that generates region proposals with various scales and aspect ratios. It scans the proposed regions to assess whether future inference needs to be carried out. The content of the proposed regions, defined by a bounding box, is classified in the second stage and the box coordinates are adjusted. We used Faster R-CNN with the backbone of residual networks (ResNet) to learn residual functions with reference to the layer inputs, instead of learning unreferenced functions. ResNet stacks residual blocks on top of each other to form a CNN. Here, ResNet50 was used, which has fifty layers of residual blocks. Residual networks are easier to optimize and gain accuracy from the increased depth of the network.

In paper [56], different YOLOv5 architectures were evaluated, finding that YOLOv5m6 with 35.7 million parameters was the optimal model to detect and classify small insect species. To improve performance and speed up training, YOLOv5m6 and Faster R-CNN with ResNet50 in our work were pre-trained on the COCO dataset [57]. We used a simple pipeline [58] with data augmentation to train the Faster R-CNN model. The data augmentation includes random vertical and horizontal image flips, image rotation, and different types of blurring. Images were re-sized to 1280 × 720 pixels for training with the two evaluated networks, and transfer learning (COCO) was used to fine-tune the parameters of the CNN.

### 3.3. Evaluation and Performance Metrics

To evaluate model performance, the precision, recall, and F1-score metrics were chosen. These metrics are based on true positive (TP), false positive (FP), and false negative (FN) insect detections. A detection is true positive if the predicted bounding box overlaps the labeled insect bounding box with more than 0.25 IoU (intersection over union) [59]. Since the insects were very small in the images, the annotated bounding box had a relatively high uncertainty. Therefore, a lower value was used compared with the commonly used 0.5 IoU. Precision is the metric that measures the proportion of positively predicted detections that are actually correct, given in Equation (6). As such, a high precision indicates a high number of correctly detected insects. Recall represents the model’s ability to predict the positives correctly out of actual positives, and is given in Equation (7). It measures the model’s ability to find and detect all labeled insects in the dataset. Recall and precision were used in conjunction to obtain a complete picture of the model’s ability to find all insects and detect them correctly. To balance precision and recall, we used the F1-score. The F1-score is calculated as the harmonic mean of precision and recall, given in Equation (8), and it provides a balance between the two metrics. It prioritizes the importance of detecting and classifying the insect species correctly over fitting the correct bounding box size.
(6)$$precision=\frac{TP}{FP+TP}$$
(7)$$recall=\frac{TP}{FN+TP}$$
(8)$$F1=\frac{2\cdot precision\cdot recall}{precision+recall}=\frac{2\cdot TP}{2\cdot TP+FP+FN}$$

A micro- and macro-average metric was computed for the model predictions of the selected seven different physical sites in the test dataset. The macro-average metric was computed as the average recall, precision, and F1-score for the model performance for each test site. The micro-average aggregates the contributions from all test sites to compute metrics based on the total number of TP, FP, and FN predictions.

## 4. Experiment and Results

A total of 717,311 images were recorded in the experimental period, monitoring honeybees and other insects visiting three different plant species.

### 4.1. Training and Validation

First, a trained model [56] was used to find insects in recordings from 10 different weeks and camera sites, as listed in Table 1. These predictions generated a large number of images of candidate insects, which were subsequently verified. Images with predictions were manually corrected for FPs, resulting in several images with corrected annotated insects and background images without insects. During quality checks, non-detected insects (false negatives) were found, annotated, and added to the dataset.

This dataset was used to create a final training dataset with an approximate split of 20% annotations used for validation. The training and validation dataset was manually corrected a second time based on the motion-enhanced images, and additional corrections were made. An additional 253 insects were found, an increase of 8% more annotated insects compared with the first manually corrected dataset. Two versions of the datasets were created with color and motion-enhanced images. The resulting final datasets for training and validation are listed in Table 3.

The training and validation datasets were used to train the two different object detection methods: Faster R-CNN with ResNet50 and YOLOv5. The models were trained with color and motion-enhanced datasets as listed below:Faster R-CNN with color images.Faster R-CNN with MIE.YOLOv5 with color images.YOLOv5 with MIE.

Each model and dataset combination was trained five times. The highest validation F1-score was used to select the best five models without overfitting the network. For each of the five trained models, the precision, recall, F1-score, and average precision (AP@.5) were calculated on the validation dataset. AP@.5 is calculated as the mean area under the precision–recall curve for a single class (insects) with an intersection over union (IoU) of 0.5. The averages of the five trained models are listed in Table 4.

The results show a high recall, precision, and F1-score for all models in the range of 85% to 92%. The trained models with motion-enhanced images have a recall 1–2% higher than with color images, but the precision is 4–5% lower. The trained YOLOv5 models have approximately a 1% higher F1-score and 2% higher AP@.5 than Faster R-CNN. Based on the results, training with motion-enhanced images does not improve the F1-score.

Training YOLOv5 and Faster R-CNN models took 18–24 h on an Intel i7 CPU 3.6 GHz and a GPU NVIDIA GeForce RTX 3090 with 24 GByte memory. Processing all camera images from the two months, shown in Figure 4, took approximately 48 h.

### 4.2. Test Results and Discussion

The test dataset was created from seven different sites and weeks not included in the training and validation datasets. A separate YOLOv5 model was trained on the training and validation dataset described in Section 4.1. This model performed inference on the selected seven sites and weeks of recordings. The results were manually evaluated, removing false predictions and searching for non-detected insects in more than 100,000 images. In total, 5737 insects were found and annotated in this first part of the iterative semi-automated process.

In the second part, two additional object detection models with Faster R-CNN and YOLOv5 were trained with motion-enhanced images. These two models performed inference on the seven sites, and predictions were compared with the first part of annotated images, resulting in the finding of an additional 619 insects. The complete test dataset is listed in Table 2.

The test dataset contains sites with varying numbers of insects, ranging from a ratio of 1.2% to 15.3% insects compared with the number of recorded images. An average ratio of 6.2% insects was found in 102,649 images. Most of the annotated insects were honeybees, but a small number of hoverflies were found at camera site S1-1. The monitoring site S1-0 (sea rocket) contained other animals such as spiders, beetles, and butterflies. Many of the images at site S1-1 were out of focus, caused by a very short camera distance to the red clover plants. Sites S2-0 and S2-1 monitored common mallow, which was not part of the training and validation dataset. Site S4-0 had a longer camera distance to the red clover plants, where many honeybees were only barely visible. In general, many insects were partly visible due to occlusion by leaves or flowers, where only the head or abdomen of the honeybee could be seen. Additional illustrations of insect annotations and detections are included in Appendix A.

In Table 5, the recall, precision, and F1-score are shown, calculated as an average of the five trained Faster R-CNN models evaluated on the seven test sites. The Faster R-CNN models were evaluated on color and motion-enhanced images. The recall, precision, and F1-score increased for all seven test sites with Faster R-CNN models trained with motion-enhanced images. The micro-average recall was increased by 15% and precision by nearly 40%, indicating that our proposed method has a huge impact on detecting small insects. This was further verified on a test dataset with a marginal distribution other than for the training and validation dataset. The F1-score was increased by 24% from 0.320 to 0.555. The most difficult test site for the models to predict was S1-0, which had a low ratio of insects (1.2%) and contained animals such as spiders and beetles not present in the training dataset.

In Table 6, the recall, precision, and F1-score are shown, calculated as an average of five trained YOLOv5 models evaluated on the seven test sites. The YOLOv5 models were evaluated on color images and motion-enhanced images. The micro-average recall was increased by 28.2% and precision by only 7%. However, the micro-average F1-score was increased by 22% from 0.490 to 0.713, indicating that motion-enhanced images did increase the ability to detect insects in the test dataset. The YOLOv5 models outperformed the Faster R-CNN trained models, achieving an increase of 16% for the micro-average F1-score from 0.555 to 0.713.

Note that camera sites S2-0 and S2-1 (with common mallow), which were not included in the training set, performed extremely well with motion-enhanced images, achieving F1-scores of 0.643 and 0.618, respectively. This indicates that the dataset for training was sufficiently varied for models to detect insects in new environments. Camera site S1-1 with red clover had a lower F1-score than other sites with the same plant (S3-0, S4-0, and S4-1). This could be related to the foreground defocus due to the close camera distance to the plants. Camera sites S3-0, S4-0, and S4-1 had the best recall, precision, and F1-score. This is probably due to the high insect ratio of 4.6–15.3% and because red clover plants were heavily represented in the training dataset.

The box plot of the F1-scores shown in Figure 5 indicates an increased F1-score with motion-trained models. It also shows a lower variation in the ability to detect insects between the seven different test sites, indicating a more robust detector.

Figure 4 shows the abundance of insects detected with two YOLOv5 models trained on color and motion-enhanced images over the two months of the experiment, including images from training, validation, and test datasets. False insect detections were typically found in the same spatial position of the image. A honeybee visit within the camera view typically had a duration of less than 120 s, as documented in [28]. A filter was therefore used to remove detections for the same spatial position within two minutes in the time-lapse image sequence. Figure 4a shows the abundance of a YOLOv5 model trained with color images. There are periods with a high difference in the filtered and non-filtered detections, probably due to a high number of false insect detections. Figure 4b shows the abundance of a YOLOv5 model trained with motion-enhanced images. The model trained with motion-enhanced images showed in general a higher number of detections than the model trained with color images, indicating more insects were found and detected. A visual overview of the results showing the micro-average F1-score for six different sites is shown in Figure 6. Here, it is evident that MIE improves the ability to detect small insects with a variety of background plants, camera views, and distances. It can also be seen that, with a higher ratio of insects, the overall F1-score is increased. Trained models with MIE are especially better at detecting insects on sites with sparse insects (Rocket top 1.2) and plants out of focus close to the camera (Clover top 2.3).

#### Challenges in Camera Monitoring

Automated camera monitoring of insects visiting flowering plants is a particularly exciting prospect for non-invasive monitoring of insects and other small organisms in their natural environment. Compared with traditional manual sampling methods, camera recording also has challenges. Many cameras are required to monitor a large area, which produces an immense amount of images for offline processing. Many remote nature locations where the system needs to operate without human intervention do not have power or mobile network coverage, such as the time-lapse cameras we installed and operated in East Greenland [60].

For insect pollinators, flowering plants are used to attract the insects; however, this requires adjustment of the camera position during the flowing season to record a high abundance of insects, and this approach is difficult to standardize. Here, the challenge was to ensure that insects were visible in the camera view during monitoring. Cameras were moved to different viewing positions to ensure blooming flowers during the monitoring period of our experiment. This is probably the most important limitation for automated camera insect monitoring to ensure a high amount of insect detections.

Calibration can also be a challenge when the camera is moved and plants grow during the monitoring period. In our experiment, we used autofocus, which often focuses on the vegetation in the background and not flowers close to the camera, which the insects frequently visit. In [56], manual focus was used to focus on flowering *Sedum* plants in the foreground. However, this only works well with vegetation that has a near-constant height during the monitoring period.

Due to the small size of the objects of interest, deep learning models will often falsely identify elements in the complex background of plants as the object of interest. This is a challenge for processing and especially for monitoring areas with a low abundance of insects, since the signal-to-noise ratio in terms of TP relative to FP detections will be very low. However, we have addressed this challenge in our proposed method, which is able to increase both the recall and precision in insect detection.

## 5. Conclusions

This work provides a public benchmark dataset of annotated insects for time-lapse monitoring from seven different sites. The dataset meets an important demand for future research in detecting small objects [34]. The test dataset includes 6356 annotated insects in 102,649 images of complex scenes of the natural environment, including three different vegetation plants. A training and validation dataset was also published and verified with our newly proposed method to train deep learning models with motion-enhanced images.

The hypothesis that motion-informed enhancement will improve insect detection in wildlife environments has been proven. The trained CNN object detectors with YOLOv5 and Faster R-CNN demonstrate a micro-average F1-score of 0.71 and 0.56, respectively, on the test datasets. This is a higher F1-score compared with the scores for the models trained on normal color images; only 0.49 was achieved with YOLOV5, and 0.32 with Faster R-CNN. Both models trained with motion-enhanced images have a higher recall than those trained with color images; YOLOv5’s and Faster R-CNN’s recall scores increased by 28% and 15%, respectively.

Our work provides a step forward for automating flying insect monitoring in a complex and dynamic natural environment using time-lapse cameras and deep learning. The study shows that more insects will be accurately detected by using motion-informed enhancement compared with state-of-the-art deep learning object detectors. The prospect of applying time-lapse photography and deep learning enables studies in a wider temporal and spatial space compared with traditional insect sampling methods, where our presented method improves the detection of small insects.

## Figures and Tables

**Figure 2 sensors-23-07242-f002:**
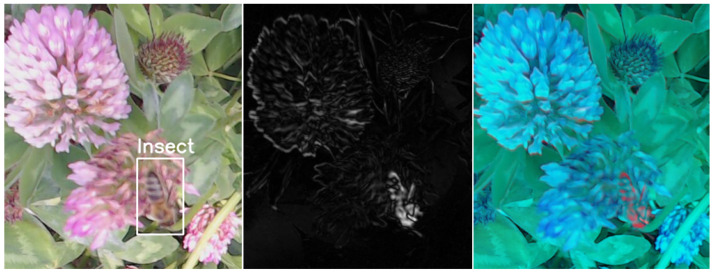
Left image shows the original colored image at time *k* with a honeybee. The center image shows how the motion likelihood ($L^{3}$) is emphasized in the image. The right image shows the motion-enhanced image (*M*), with the red indicating information about the moving insect.

**Figure 3 sensors-23-07242-f003:**
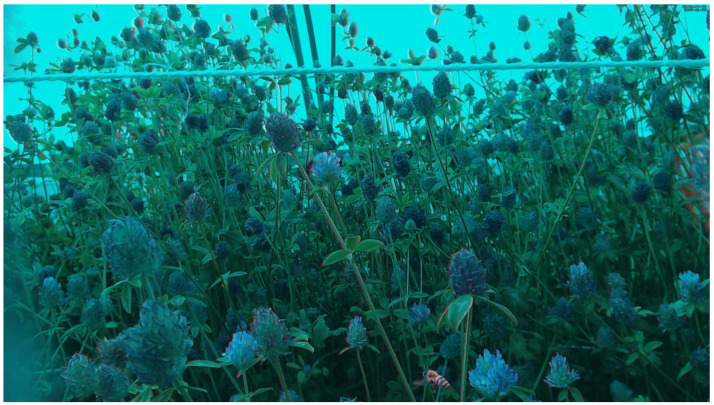
A full-scale 1920 × 1080 motion-enhanced image with one honeybee.

**Figure 4 sensors-23-07242-f004:**
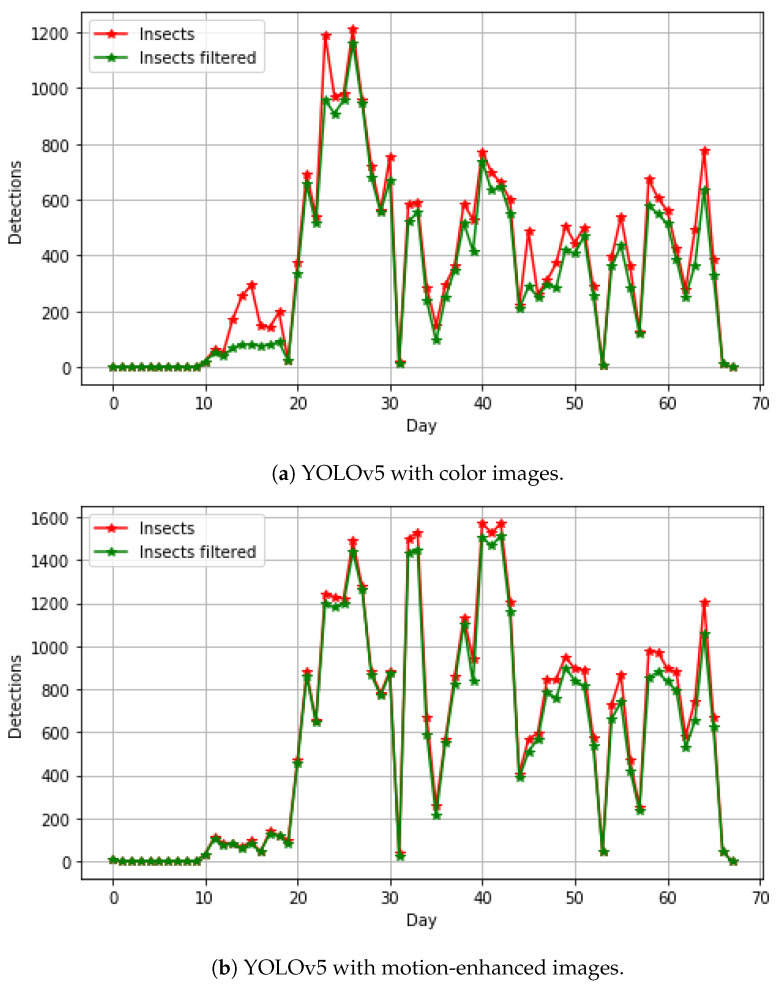
The abundance of insects from the two-month monitoring period of flowers and insects. A two minute filter is used to remove detections at the same spatial position in the time-lapse image sequence. The red and green curves show the non-filtered and filtered detections, respectively. The difference between the curves indicates false predictions or an insect detected at the same position within two minutes.

**Figure 5 sensors-23-07242-f005:**
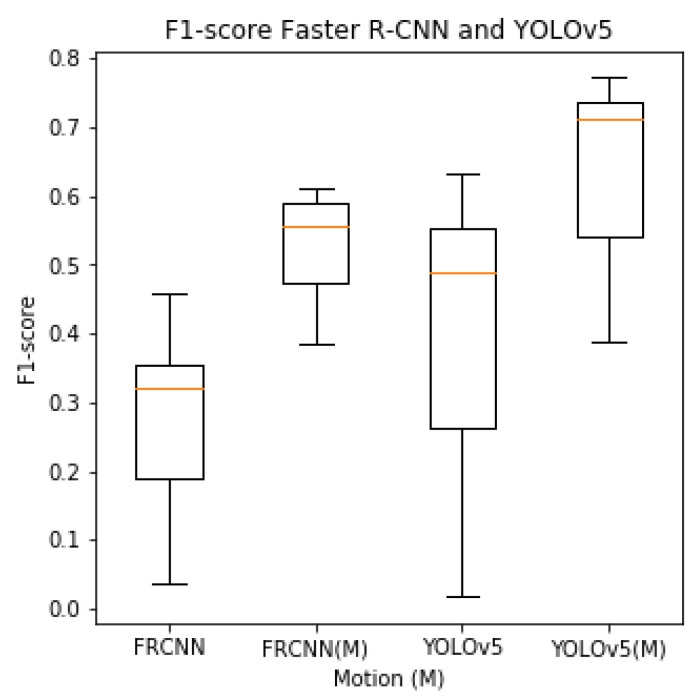
Box plot of F1-score for seven sites of YOLOv5 and Faster R-CNN models trained with color and motion-enhanced (M) images. The horizontal orange mark indicates the micro-average F1-score based on all seven test sites.

**Figure 6 sensors-23-07242-f006:**
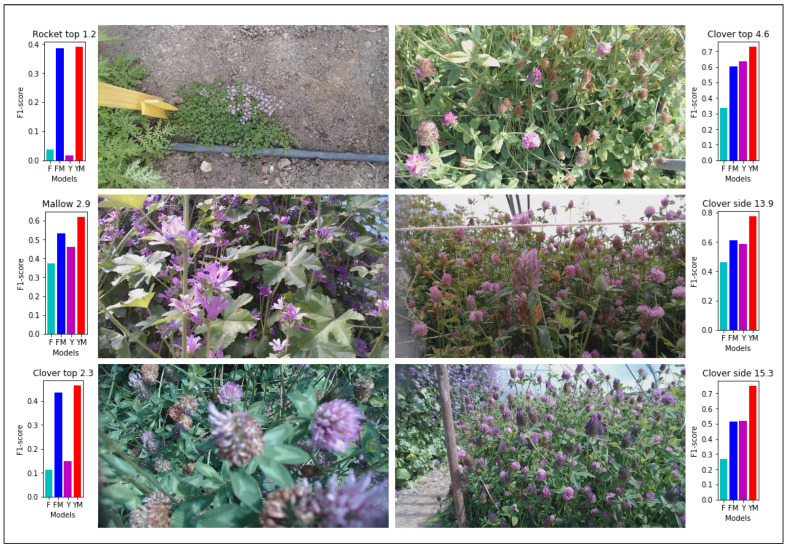
Images with micro-average F1-scores from six different sites for YOLOv5 and Faster R-CNN trained models. Faster R-CNN (F)—light blue, Faster R-CNN with motion (FM)—blue, YOLOv5 (Y)—purple, and YOLOv5 with motion (YM)—red. The six sites are Rocket top 1.2 (S1-0), Mallow 2.9 (S2-1), Clover top 2.3 (S1-1), Clover top 4.6 (S4-1), Clover side 13.9 (S3-0), and Clover side 15.2 (S4-0).

**Table 2 sensors-23-07242-t002:** Test dataset with the number of annotated insects in recordings from seven different camera sites and weeks. System number and camera Id (Sx-0/1) identify each camera. The percentage ratio of annotated insects relative to the number of images recorded during each week is shown. The average ratio is 6.2% insects, based on 6356 annotations in 102,649 time-lapse recorded images. The flowering plants were observed with a top or side camera view. The plant species are sea rocket (*Cakile maritima*), red clover (*Trifolium prantese*), and common mallow (*Malva sylvestris*).

Cam.	Week	Insects	Images	Ratio (%)	View	Plant
S1-0	24	170	14,092	1.2	Top	Rocket
S1-1	29	333	15,120	2.2	Top	Clover
S2-0	24	322	14,066	2.3	Side	Mallow
S2-1	26	411	14,011	2.9	Side	Mallow
S3-0	28	2100	15,120	13.9	Side	Clover
S4-0	27	2319	15,120	15.3	Side	Clover
S4-1	30	701	15,120	4.6	Top	Clover

**Table 3 sensors-23-07242-t003:** Final training and validation datasets with annotated insects and number of images. Background is the number of images without any insects.

Dataset	Insects	Images	Background
Train	2499	3783	1953
Validate	568	946	508
Total	3067	4729	2461

**Table 4 sensors-23-07242-t004:** Average validation recall, precision, F1-score, and AP@.5 for five trained Faster R-CNN and YOLOv5 models with color and motion-enhanced images.

Model	Dataset	Recall	Prec.	*F*1-Score	AP@.5
FR-CNN	Color	0.867	0.889	0.878	0.890
FR-CNN	Motion	0.889	0.862	0.875	0.900
YOLOv5	Color	0.888	0.897	0.892	0.914
YOLOv5	Motion	0.919	0.852	0.884	0.924

**Table 5 sensors-23-07242-t005:** Average recall, precision, and F1-score for each camera site used in the test dataset. The average was calculated based on five trained Faster R-CNN with ResNet50 models compared with five models trained with motion-enhanced images. The macro- and micro-average metrics cover results from all seven camera sites and weeks. The best macro- and micro-average of recall, precision, and F1-Score are marked with bold.

	FR-CNN	Motion	FR-CNN	Motion	FR-CNN	Motion
Camera	Recall	Recall	Precision	Precision	*F*1-Score	*F*1-Score
S1-0	0.051	0.262	0.032	0.758	0.037	0.385
S1-1	0.141	0.413	0.112	0.488	0.112	0.435
S2-0	0.305	0.529	0.250	0.650	0.274	0.576
S2-1	0.355	0.496	0.398	0.599	0.374	0.532
S3-0	0.404	0.487	0.538	0.840	0.459	0.612
S4-0	0.178	0.365	0.539	0.891	0.267	0.515
S4-1	0.496	0.585	0.262	0.634	0.337	0.603
Macro	0.276	**0.448**	0.305	**0.694**	0.266	**0.522**
Micro	0.300	**0.446**	0.344	**0.751**	0.320	**0.555**

**Table 6 sensors-23-07242-t006:** Average recall, precision, and F1-score for each camera site used in the test dataset. The average was calculated based on five trained YOLOv5 models compared with five models trained with motion-enhanced images. The macro- and micro-average metrics cover results from all seven camera sites and weeks. The best macro- and micro-average of recall, precision, and F1-Score are marked with bold.

	YOLOv5	Motion	YOLOv5	Motion	YOLOv5	Motion
Camera	Recall	Recall	Precision	Precision	*F*1-Score	*F*1-Score
S1-0	0.028	0.284	0.019	0.693	0.017	0.389
S1-1	0.126	0.502	0.210	0.437	0.147	0.463
S2-0	0.288	0.630	0.619	0.674	0.376	0.643
S2-1	0.335	0.635	0.784	0.621	0.461	0.618
S3-0	0.442	0.694	0.890	0.879	0.587	0.772
S4-0	0.368	0.665	0.890	0.865	0.517	0.747
S4-1	0.486	0.733	0.917	0.727	0.634	0.727
Macro	0.296	**0.592**	0.619	**0.699**	0.392	**0.623**
Micro	0.377	**0.659**	0.718	**0.784**	0.490	**0.713**

## Data Availability

The dataset is published at: https://vision.eng.au.dk/mie (accessed on 10 August 2023).

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
