# Peer review of "Object Detection of Small Insects in Time-Lapse Camera Recordings"

_sensors, 2023, doi:10.3390/s23167242_

Round 1

Reviewer 1 Report

This paper's main contribution: First, provide a dataset of primary honeybees, Second, present a detecting insects pipeline method. Some content needs to be modified by the author.

Introduction: When discussing the challenges and methods of small object detection, it would be beneficial to further expand on the existing research, particularly on the techniques and approaches for detecting small objects in low-resolution remote sensing images. Providing more specific details and examples can help readers better understand the background of relevant studies.

Related Work: When discussing the related research, it would be helpful to provide a more comprehensive review, such as highlighting the advantages and limitations of different research methods, important findings, or future research directions. Additionally, regarding your statement in the article about annotated datasets being essential for data-driven insect detectors, you can provide a description or critique of existing datasets to emphasize the significance of the insect dataset you have provided for small object detection. 

Experiments and Results: It would be valuable to compare the performance of the models under different conditions and datasets and analyze the reasons for differences in results. For example, for the S1-1 point illustrated in Figure 6, the severe foreground defocusing issue due to the close camera distance to the plants, can improvements be made in future work to further enhance the detection performance of the model under similar conditions? 

Conclusion: In summarizing the paper, the author can further elaborate on the prospects of applying time-lapse photography and deep learning in this field, as well as the importance of this study in filling the existing research gap.

Reviewer 2 Report

The content of this paper was to report a novel technique for small Insects in a time-lapse camera. It provides useful information and new technique. However, the content of the paper needs to be enhanced to make it more readable.

1. Please delete some pictures. Too many pictures only make it too complicated to read.

2. Please rewrite the section “4.2. Object detection with deep learning”. It was very difficult to understand the methods used in this study, especially Faster R-CNN, YOLOv5, etc. Please describe each method briefly, not only by citation.

3. What are the criteria to evaluate the performance of data analysis? Why these criteria were selected?

4. What is the requirement or criterion of the user (farmer or entomologist)? F1-score is the criterion for the data analyzer, not for users.

Moderate editing of English language required

Round 2

Reviewer 2 Report

The content of the revised version has improved significantly.

Minor editing of English language required